# Epidermal Club Cells in Fishes: A Case for Ecoimmunological Analysis

**DOI:** 10.3390/ijms22031440

**Published:** 2021-02-01

**Authors:** Sumali Pandey, Craig A. Stockwell, Madison R. Snider, Brian D. Wisenden

**Affiliations:** 1Biosciences Department, Minnesota State University Moorhead, Moorhead, MN 56563, USA; sumali.pandey@mnstate.edu; 2Department of Biological Sciences, North Dakota State University, Fargo, ND 58105, USA; craig.stockwell@ndsu.edu (C.A.S.); mrsnides5@gmail.com (M.R.S.); 3Environmental and Conservation Sciences Graduate Program, North Dakota State University, Fargo, ND 58105, USA

**Keywords:** mucosal immune system, epidermal club cells, Ostariophysi, ecoimmunology

## Abstract

Epidermal club cells (ECCs), along with mucus cells, are present in the skin of many fishes, particularly in the well-studied Ostariophysan family Cyprinidae. Most ECC-associated literature has focused on the potential role of ECCs as a component of chemical alarm cues released passively when a predator damages the skin of its prey, alerting nearby prey to the presence of an active predator. Because this warning system is maintained by receiver-side selection (senders are eaten), there is want of a mechanism to confer fitness benefits to the individual that invests in ECCs to explain their evolutionary origin and maintenance in this speciose group of fishes. In an attempt to understand the fitness benefits that accrue from investment in ECCs, we reviewed the phylogenetic distribution of ECCs and their histochemical properties. ECCs are found in various forms in all teleost superorders and in the chondrostei inferring either early or multiple independent origins over evolutionary time. We noted that ECCs respond to several environmental stressors/immunomodulators including parasites and pathogens, are suppressed by immunomodulators such as testosterone and cortisol, and their density covaries with food ration, demonstrating a dynamic metabolic cost to maintaining these cells. ECC density varies widely among and within fish populations, suggesting that ECCs may be a convenient tool with which to assay ecoimmunological tradeoffs between immune stress and foraging activity, reproductive state, and predator–prey interactions. Here, we review the case for ECC immune function, immune functions in fishes generally, and encourage future work describing the precise role of ECCs in the immune system and life history evolution in fishes.

## 1. Introduction

Epidermal club cells (ECCs) have been extensively studied in the context predator–prey ecology, because they are the presumed source of chemical alarm cues released during predator attacks [1,2]. Von Frisch was the first to report observations of antipredator behavior in minnows in response to water-soluble compounds released from damaged tissues of an injured conspecific [1,2], and that only injured epidermal tissue produces these behavioral responses [3]. These observations stimulated research to survey species with similar behavioral responses. Pfeiffer published a review [4] that included much of his own research, showing that alarm reactions were widespread among fish species in the superorder Ostariophysi, and absent in the non-Ostariophysans tested. He also noted that ECCs were unique to the Ostariophysi and concluded that these club cells were a strong candidate for the source of the alarm cue. He labeled the cells “alarm substance cells”, arguing that ECCs, being on the surface of the body, thin walled, and having no duct with which to release their contents to the external environment, ECCs would be among the first cells ruptured in an attack by a predator and release of their contents would thereby indicate the presence of an actively foraging predator. Thus, it seemed as if ECCs contained a chemical alarm signal, or alarm pheromone, which warned conspecifics of the presence of danger [4].

Evolutionary ecologists noted a flaw in the argument for the evolutionary maintenance of ECCs as the source of an alarm pheromone [5,6]. Although injury-released compounds from damaged epidermis provide great benefits to nearby conspecifics that receive and use that information, an individual fish would not realize a fitness benefit for investing in ECCs and thus their maintenance must be explained by some other adaptive function, which benefits the sender. Smith [5] hypothesized that senders may benefit from their own injury-released compounds if alarm cues attracted additional predators which in turn increased the prey item’s survival probability [7,8]. Thus, in these specific cases, ECCs may be considered exaptations [9]. However, ECCs have a broad phylogenetic distribution; thus, these highly specified hypotheses posited by Smith are not likely to explain the evolutionary origin and maintenance of club cells in the thousands of fish species that possess them.

Because rupture of ECCs is correlated with predation/parasitism events, there is strong selection on receivers to detect and recognize constituents of ECCs as indicators of risk, and consequently execute appropriate anti-predator [10] or anti-parasitic behaviors [11,12,13]. Because behavioral alarm reactions are maintained by receiver-side selection, the compounds released are correctly considered as cues (public information), not as signals (by definition, a signal requires a benefit to the sender [14]). The previous label of “alarm pheromone” (a type of signal) is misleading because it confuses the evolutionary understanding of the origin and function of ECCs [15,16].

In the 43 years since Pfeiffer [4], the diversity of fishes tested for alarm reactions to conspecific skin has been broadened significantly, and we now know that most fish species generally exhibit antipredator responses to compounds released from injured specifics [10,17]. In fact, most aquatic organisms from Platyhelminthes, Arthropoda, Mollusca, to Amphibia have similar responses. Notably, few of these other groups of aquatic organisms possess specialized structures analogous to epidermal club cells, undercutting a requirement for specialized structures for production of alarm cues [10,18]. The case for an alarm function for epidermal club cells in ostariophysan fishes is further undermined by data showing no reduction in cue potency when ECCs have been suppressed [19,20] or are absent [21], suggesting that ECCs may be a contributor to, but not the sole source of, the alarm cue.

An alternative hypothesis for the function for ECCs is that they have a role in immune defense [22,23,24,25]. The epidermis is a natural barrier to pathogens and environmental insults of various kinds [26]. If club cells have a primary role in immune function, which benefit the individual that produces them, then over evolutionary time ECCs could have acquired an incidental role as a contributor to species-specific odor signatures recognized by conspecifics as indicators of danger.

In this review, we discuss the basic biology and distribution of ECCs, the existing evidence for immunological function of club cells, and propose that ECCs provide a convenient tool for ecoimmunological studies to investigate interactions among immune function and trade-offs with other ecological functions such as predator avoidance and reproduction.

## 2. Epidermal Club Cells in Fish

### 2.1. Phylogenetic Distribution of Epidermal Club Cells

Epidermal club cells (ECCs) have now been reported in five superorders of teleosts and in chondrostians, i.e., including many groups beyond what Pfeiffer [4] originally reported (Table 1). Epidermal skein cells of lamprey (superclass Agnatha) are distinct and non-homologous with club cells in Osteichthyes (Table 1). To our knowledge, a systematic survey for the presence of ECCs among major fish groups has not yet been conducted since Pfeiffer [4]. In the current review, as in Pfeiffer [4], the apparent absence of ECCs in many groups is likely an artifact of low sampling effort. Snider [27] recently reported that club cell prevalence within a sample of 28 fish was 11%, meaning that numerous individuals would need to be sampled to definitively confirm the absence of ECCs. Thus, case studies reporting an absence of ECCs should be evaluated critically to assure that a sufficient sample size of individuals were surveyed. Even with limited sampling of fish taxa, ECCs have been observed in five superorders, suggesting that either ECCs are ancestral to stem actinopterygians, or they have been innovated independently in multiple lineages, perhaps in response to a common and ubiquitous component of fish biology, such as immune defense.

The superorder ostariophysi is conspicuous for its diversity: Almost 8000 species, in 77 extant families [28], and virtually all have well-developed ECCs, leading Fink and Fink [29] to conclude that ECCs are a synapomorphic trait shared by all members of the Ostariophysi. A conspicuous exception is found in the weakly electric fishes of the order Gymnotiformes, which lack ECCs even though they descended from ancestors that possessed them [29]. Ostariophysans occur exclusively in freshwater, and collectively comprise about 28% of the world’s fish species [28].

ECCs in non-Ostariophysan groups have received very little attention from ecologists, immunologists, or histologists. Smith [5] summarized earlier work on darters (order Perciformes). Since then, there have been additional data on ECCs in other percids: Walleye [30] and yellow perch [25]. All percids examined thus far have abundant, large ECCs despite being listed by Pfeiffer [4] as lacking them (Table 1). Fish in the order Cyprinodontiformes (“tooth carps” such as killifish, pupfish, springfish, poolfish, and poeciliids including guppies, swordtails, and mosquitofish) are ecologically similar to many ostariophysan species in being small-bodied, open-water obligate shoaling fishes. Recent research on two species of pupfish *C. tularosa and C. nevandensis amargosa* as well as Pahrump poolfish *Empetrichthys latos* and White River springfish *Crenichthys baileyi moapae* showed low density of ECCs (relative to cyprinids such as fathead minnows and zebrafish; Table 1), with many individuals lacking them completely [27]. In fact, *C. b. moapae* lacked ECCs, but only 10 individuals were sampled [27]. This comparative approach raises intriguing questions about the covariance structure among phylogenetic history, ecology of predator–prey interactions, host–pathogen interactions, potential for range expansion (invasive species), and other environmental variables. An ecoimmunological approach would address many of these questions.

### 2.2. Histochemical Characteristics of ECCs

Fish skin comprises three layers: The mucous layer, the epidermis, and the dermis. The epidermis of ostariophysian fishes contains four cell types: Epidermal, mucus, granular, and club cells. Amongst these, ECCs are identified in histological sections as relatively large, sometimes binucleate cells (notice the cell in Figure 1A) in the mid-epidermal layer [26], which remain unstained with periodic acid–Schiff (PAS), hematoxylin, or eosin stains (Figure 1). With periodic acid–Schiff stain, club cells do not sequester the stain, suggesting a lack of carbohydrate content [63]. Unlike typical ostariophysans, ECCs in eels have a secretory vacuole [26]. In carp, club cells are significantly larger than mucus cells, about 27 × 23 µm in diameter, with an indented nucleus, located centrally, surrounded by electron-dense cytoplasmic structures [52]. The cell periphery contains a wide belt of electron-lucent cytoplasm containing contorted microfilaments, associated with the desmosomes, about 250 nm long and 10 nm in diameter [22,64]. These filaments are arranged randomly in ostariophysans but have a uniform distribution in eels [65]. The club cells in the skin of pupfish and poolfish appear near the surface, and the nucleus is positioned at the base of the cell (Figure 1).

Because ECCs are a likely contributor to chemical alarm cues, attempts at biochemical characterization of the biologically active compounds that induce alarm reactions may provide clues to ECC function. Biochemical characterization of alarm substance began in the 1940s and involved solvent extraction and chromatography-based methods [66]. The substances proposed to be present in the alarm cue include purine- and pterin-like substances [67], including ichthyopterin, isoxanthopterin [68], hypoxanthine-(3N)-oxide [69], chondroitin sulfate [70], toxins and pharmacologically active compounds [71], proteins [18,72], and bacteria [73]. The ability of many of these substances to induce alarm behavior has been investigated [73,74,75,76]. While low molecular weight (330–550 Da) substances did not induce an alarm behavior, substances of high molecular weight (>1500 Da) did [76]. In a non-invasive procedure, cell-free media obtained from primary culture of skin cells scraped from fish skin induced darting behavior in creek chub, indicating that injury or blood components are perhaps not required for this behavior [77]. Skin extract derived from young fathead minnows before the skin contains ECCs also induces alarm reactions [21]. Indeed, a detailed study of chemical composition of alarm cue is an active area of investigation, and the general consensus is that alarm cues present in skin extract, at least in minnows, contains more than one active component, including bacteria, and full potency to induce a behavioral response requires all components to be present [16,70,73]. Characterization based on whole skin extract includes many epidermal components in addition to ECCs, therefore the cellular source of these substances remains unclear.

With the lack of a specific cell surface marker for club cells, studies for club cell specific content have relied on observations based on histology and immunohistochemistry. These studies have identified chondroitin sulfate, keratin sulfate [41], lethal factor toxin [43,61], serotonin [44], and calcium binding proteins [78,79] inside the club cells. The presence of these well-characterized substances in club cells suggest a role for these cells in other biological processes. Several biological functions have been ascribed to the substances found inside club cells. For example, chondroitin sulfate and keratin sulfate are components of extracellular matrix and may function as immunomodulators or in organismal development [80,81,82,83,84]. Serotonin is a neurotransmitter, which may function as an environmental sensor in the skin [85,86]. Calcium-binding proteins serve to transport Ca^2+^, in buffering and enzymatic systems, in cytoskeletal organization, cell motility, and differentiation [87,88,89]. The biological role of these substances has been studied in various systems; however, their role in the context of ECCs remains uninvestigated.

## 3. An Overview of the Immune System in Fishes

Fishes colonize diverse environments, including deep sea, polar regions, freshwater, and marine ecosystems, and exhibit amazing physiological adaptations to accommodate varying levels of salinity, temperature, alkalinity, and light. These varied environments expose fish’s immune system to numerous environmental stressors. Increased stress in varied environments can elevate plasma cortisol levels in fish [90], which can further influence their immune system [91,92]. These varied environments can also influence parasite risk as well as the microbiome, which can further influence fish’s immune system [93]. For example, salinity levels mediate the distribution of various gastropods and their associated digenes that parasitize fishes [94,95]. Exposure to these infectious disease agents provides a robust selection pressure on the fish defense system.

The immune system of fish has garnered significant attention lately because: (1) Most immunological studies relevant to human health are carried out in mice or humans. For evolutionary biologists, fish provide a critical comparative group with which to study immune system evolution; (2) fish, such as zebrafish, serve as an excellent model for immunological investigations due to their genetic and physiological similarities with humans, and due to the expansion of optical, genetic, and chemical investigative tools [96]; (3) understanding fish disease is obviously relevant for management of fish populations that are reared for food, sport, or commercial fisheries, and for ecosystem health.

Amongst fish, the immune system of teleosts, which possesses elements of innate and adaptive immune system, is most well-studied. A critical function of the immune system is to distinguish the self from the non/altered self. In multicellular organisms, the immune system is an interconnected and interacting system comprising numerous macromolecules, cells, tissues, and organs, and is broadly classified as an innate and adaptive immune system.

The response to a pathogenic threat is immediate and faster for the innate immune system compared to the adaptive immune system. Innate host defense mechanisms are found in nearly all living organisms, including unicellular prokaryotes (e.g., clustered regularly interspaced palindromic repeats (CRIPRs)), metazoans, and protozoans. The innate immune system is believed to have arisen ~1000 million years ago, and specific components of the adaptive immune system were present when jawed vertebrates first appeared about 450 million years ago [97,98].

Adaptive immunity is activated subsequent to innate immunity if the antigen persists. The two main cell types that mediate adaptive immune responses are T and B lymphocytes. While jawless fish only possess T- and B-like cells, gnathostomes have T and B lymphocytes—most similar to the ones found in mammals [99,100]. Components of the adaptive immune system are relatively slow responders compared to innate immune system components. This delay in response is due, in part, to the requirement that antigen capture and presentation by dendritic cells must occur before lymphocytes are activated.

The next section describes the innate and adaptive immune system in teleost fish and concludes with a discussion of known and proposed immunological functions of club cells.

### 3.1. Innate Immune System in Fish

Innate immune components provide defense against invading pathogens in several ways: The first line of defense is provided by the physical barrier that prevents the entry of a pathogen. In fish, this includes the skin, gut, gills, and the olfactory organ [101]. Of relevance to this review is skin-associated lymphoid tissue (SALT) [102]. In addition to providing a physical barrier, these external surfaces are lined with mucus-producing goblet cells that entrap invading pathogens. The main components in fish mucus include mucins, enzymes (e.g., lysozymes, acid and alkaline phosphatases, cathepsins, esterases, etc.), proteases, antimicrobial peptides, lectins, secreted immunoglobulins (predominantly IgT), and other proteins (e.g., lactoferrin, histones etc.), which can provide a strong chemical defense against pathogens [102].

Besides mucus-producing goblet cells, the cellular components of innate immunity in fish comprise tissue resident cells, such as dendritic cells, mast cells, and recruited cells such as monocytes/macrophages, neutrophils, and natural killer cells. These cells are produced in the primary lymphoid organs of fish, namely the head kidney and thymus [100]. Dendritic cell subsets in teleosts, as in mammals, function as professional antigen-presenting cells, thereby bridging innate and adaptive immune systems [103]. Innate immune cells are activated in response to foreign ligands called pathogen-associated molecular patterns (PAMPs) or damage-associated molecular patterns (DAMPs) through germline-encoded pattern recognition receptors (PRRs). Several categories of PRRs have been identified in fish, including a diverse array of twenty plus Toll-like receptors (TLRs) [104], nucleotide-binding domain, leucine-rich repeat-containing proteins (NLRs), retinoic acid inducible gene I-like receptors (RLRs) [105] and novel immune-type receptors (NITRs), diverse immunoglobulin domain-containing proteins (DICPs), polymeric immunoglobulin receptor-like proteins (PIGRLPs), novel immunoglobulin-like transcripts (NILTs), and leukocyte immune-type receptors (LITRs) [106]. Once activated, these innate immune cells carry out several roles to eliminate an antigen—including degranulation, phagocytosis, secretion of cytokines and chemokines to activate and/or recruit other leukocytes to the site of action.

The cell-free, soluble mediators of innate immunity include serum proteins such as complement proteins and acute phase proteins. The complement pathway encompasses a proteolytic cascade, which upon activation, works to lyse the pathogen, or opsonization to facilitate phagocytosis of the pathogen or recruit other leukocytes (inflammation) to the site of action. There are three typical pathways by which the complement proteins’s proteolytic cascade can be activated: The classical pathway (antigen-antibody complex mediated), alternative pathway (spontaneous hydrolysis of complement proteins on the pathogen’s surface), and lectin pathway (includes the engagement of mannose-binding lectin, a type of pattern recognition receptor, with its cognate ligand—mannose). While there are species-specific differences in fish, most of the mammalian complement proteins have homologues in teleost species [107]. In addition to complement proteins, acute phase proteins, such as C-reactive protein and serum amyloid protein play an important role in early inflammatory response and pathogen elimination. In mammals, these acute-phase proteins affect body temperature, vascular permeability, and bone-marrow derived cell production, thereby further amplifying inflammation. The pro-inflammatory cytokines secreted during inflammation, such as interleukin-1, -6 and tumor necrosis factor (TNF)-α, can stimulate hepatocytes to release acute-phase proteins [108].

Lastly, the concept of immunological memory is traditionally associated with the adaptive immune system. However, emerging evidence now suggests that innate immune cells can be trained to launch a heightened response against a secondary infection. Trained but unstimulated carp macrophages showed increased phagocytosis and inflammatory cytokine response [109]. Upon homologous or heterologous stimulation these macrophages responded with increased reactive oxygen species and nitric oxide levels as compared to untrained macrophages. Albeit non-specific, the heightened magnitude and kinetics of innate immune response upon reinfection is likely to involve epigenetic mechanisms [110] and could perhaps be passed on from parents to offspring [111,112,113]. Establishing trained innate immune response would especially be of significance for larval aquaculture, since larvae do not have a fully developed adaptive immune response [114,115].

### 3.2. Adaptive Immune System in Fish

The two main cell types that mediate adaptive immune responses are T and B lymphocytes. B lymphocytes secrete antibodies, which are either bound to the cell membrane or secreted. Membrane-bound antibodies function as antigen receptors (B-cell receptor) and cell signaling molecules. By contrast, secreted antibodies function to bind antigen and neutralize it or participate in antibody-dependent cell-mediated cytotoxicity or function as opsonins that activate complement pathways or mediate phagocytosis. In teleost fish, three antibody sub-types have been identified—IgM, IgD, and IgZ/T. The IgM antibody is the most conserved antibody class in form and function across vertebrates. It is the most abundant sub-type, has been identified in all jawed fish except in coelacanths, and its serum levels increase in response to infection. IgM can exist in a membrane-bound or secreted form, as a monomer or a tetramer joined by disulfide bonds (and not the J chain as in mammals) [116,117]. IgD antibody, on the other hand, is mostly found in transmembrane form. Secreted IgD has been detected only in catfish (*Ictalurus punctatus*) [118] and rainbow trout (*Oncorhynchus mykiss*) [119], and its function in teleosts remains relatively less clear. The third class of immunoglobulins, IgZ/T, is believed to be functionally homologous to mammalian IgA and predominates in mucosal secretions, including teleost gills, gut, and skin [120,121,122]. In the skin, they were found to coat the majority of the skin microbiota [123].

In addition to being the primary antibody secreting cells, B lymphocytes are also capable of phagocytosis [124]. Phagocytosis involves internalization of solid particles (including microbial pathogens) into cytoplasmic vesicles called phagosomes, which then mature into antimicrobial vesicles called phagolysosomes. Within the phagolysosomes, the pathogen is degraded. During this process, antigens may get mounted on major histocompatibility complex-II molecules to be presented to naïve T lymphocytes for initiation of adaptive immune responses. Indeed, B lymphocytes in teleosts have also been shown to function as antigen presenting cells [125,126]. The phagocytic ability of B lymphocytes is mostly studied, thus far, in context of IgM^+^ B cell subsets [127]. In the absence of bone marrow in teleosts, progenitor B cells and plasma cells originate from and mature in the anterior (head) kidney and are activated in the posterior kidney or spleen [128]. B cell development is regulated by several transcription factors including E2A, EBF, Pax5, Blimp1, Xpb1, and Ikaros [129]. However, the factors regulating the homing of B lymphocytes are less clear.

T lymphocytes are characterized by the T cell receptor (TCR), with the help of which they recognize the antigens. Like in mammals, two classes of T lymphocytes have been identified in teleosts based on the type of TCR they carry—(i) the true T lymphocytes of the adaptive immune system, with αβ- receptors and (ii) the innate-like γδ T lymphocytes [130]. The TCR on γδ T lymphocytes in fish, like in mammals, is encoded by Vγ and Vδ gene segments, which can recognize unprocessed antigens, without major histocompatibility restriction. In fish, they played an important role in antigen-specific IgZ production in intestinal mucosae. They were also shown to phagocytose and present antigens to initiate antigen-specific CD4^+^T cell proliferation and subsequently induce B cell activation and IgM production, thus suggesting that γδ-T lymphocytes in fish can bridge innate and adaptive immunity [131]. The αβ-T lymphocytes, unlike B lymphocytes, are MHC-restricted, which means that they recognize antigens only if that antigen is mounted on an MHC molecule (class I or II) on the surface of an antigen-presenting cell. αβ-T lymphocytes are further categorized into T-helper and cytotoxic T lymphocytes. While T-helper cells utilize CD4 co-receptor that stabilizes the interaction of TCR with the antigen-presenting MHC molecule, the cytotoxic T lymphocytes utilize a CD8 co-receptor. CD4^+^ T-helper cells predominantly help other cells, such as B lymphocytes and macrophages, and CD8^+^ cytotoxic T lymphocytes kill infected and tumor cells using perforins and granzymes. Like in mammals, T-helper 1, 2, and T regulatory subsets of CD4^+^ T lymphocyte subsets have been identified in fish [132,133,134].

Lastly, immunological memory confers long-term protection against fish pathogens. Immunological memory cells are characterized as: (1) Immune cells that are maintained long after first exposure to an antigen, without perpetual antigen stimulation; (2) are antigen-specific; (3) undergo genetic changes, such as somatic hypermutation and recombination upon first exposure to the antigen, which allows them to respond more rapidly and effectively upon secondary exposure. Studies have shown that long-term protection is induced upon vaccination in fish [134,135,136]. With regards to B-lymphocytes, class-switch recombination, affinity maturation, and clonal expansion play an important role in conferring long-term protection, and these processes are aided by T-helper cells. Teleosts lack the ability to carry out class-switch recombination, even though they express activation-induced cytidine deaminase (AID), an enzyme involved in class-switch recombination [137,138]. This inability is most likely due to differences in catalytic domains of AID enzyme and in cis elements of the IgH gene [139]. Affinity maturation and clonal expansion of B lymphocytes, however, have been documented in teleosts [122,140]. While high-affinity antibodies do appear in fish, the response time of teleost IgM is much slower than in mammals [140,141]. Species diversity is also important to consider in this regard. For example, Atlantic cod *Gadus morhua* [142] and pipefish *Syngnathus typhle* [143] do not express MHC-class II molecules; thus, B-cell responses cannot rely on MHC-restricted T cell help. With regards to T cells, adoptive transfer of CD8α^+^ T lymphocytes successfully transferred protection in fish [144], and CD8+ lymphocytes increased upon transfer of peripheral blood lymphocytes in fish from vaccinated donors [145]. Nonetheless, in contrast to our understanding of immunological memory in mammals, the mechanisms underlying immunological memory in fish are less well understood. Studies to investigate the phenotype of immune memory cells, their role in mediating memory and their location in fish still would benefit from knowing the cell surface markers for memory cells and tools to investigate those markers [145]. While the concept of immunological memory has long been associated with adaptive immune system, the concept of trained immunity in the innate immune system and its role in defense against skin pathogens is relatively newer.

### 3.3. Immunological Function of Epidermal Club Cells

Teleost skin is not keratinized, and therefore, skin cells, including ECCs located in the mid-epidermis, are in close contact with the water and the surrounding environment. Due to their structural location, they are likely to function as innate immune cells in fish immune system. The effect of several environmental stressors/immunomodulators on ECCs has been investigated (Table 2). This includes alarm cues (skin extracts from same or different fish species) [25], hypoxanthine-3-N-oxide [25], cortisol [146], parasites [25,147], infections with water mold [25,148] and bacteria [149], ultraviolet radiation [19,20,24,150,151] and white-blue light [79], water pollutants such as acidity [152], salinity [152], manure [152,153], detergents [154], azo dye [155], heavy metals such as cadmium [20,25,152], lead [152], and copper [156], mechanical injury [25,152], testosterone [157], and food ration [158,159]. Some of these stressors have shown contradictory results—for example, parasite infestation does not always result in increased ECC density. While the ECC density in minnows increased when exposed to cercariae that parasitizes turtles [25], parasites specialized to evade the immune system of minnows do not provoke proliferation of ECCs [160]. Club cell density increased upon infection with water mold [25,148], but decreased upon infection with bacteria *Aeromonas hydrophila* [149]. Similarly, ECCs were sensitive to high levels of ultraviolet radiation and decreased in density in fathead minnows [19,20,150], perhaps in response to a short a cortisol response to UV, but ECCs underwent hyperplasia and hypertrophy and were found to be photoprotective in razorback suckers *Xyrauchen texanus* [24]. A recent study showed that white-blue and blue light exposure increased the expression and colocalization of calbindin and calretinin proteins in club cells, which might be associated with the photoprotective role of club cells [79]. Lastly, dissolved organic carbon was found to be protective against UV ray-mediated damage to ECCs [150].

Skin wounds can become an entry point for pathogens and may hinder fish growth, thereby resulting in huge economic losses for aquaculture [102]. Thus, understanding the immunological processes involved in wound-healing is important to design evidence-based diagnostics and therapeutics. The cascade of cutaneous wound healing involves removal of dead tissue, re-epithelization of the wound in acute phases, and reorganization of the dermal connective tissue in chronic phase [161]. While the acute phase starts immediately, the chronic phase can last for days and months, depending on the wound severity, rearing environment, overall immune, and nutritional status of a fish [162,163,164,165]. Iger et al. described ECCs in the context of experimentally wounded carp, and noted that the number of ECCs in the wound area was similar to that of normal area [52]. They also noted that ECCs were the last cells to differentiate from filament cells during the re-epithelization period. In larval ontogeny, ECCs are the last to differentiate from the filament cells [166]. In fathead minnows, ECCs first appear at about 28–37 days post-hatch [21]. In sturgeon larvae, the mucus cells were observed in week 1 larvae, but club cells did not appear until week 4 [167]. The late appearance of ECCs, both during ontogenesis and after wounding, possibly indicates their reduced protective role during wound healing, compared with filament or mucus cells. For example, phagocytic activity was reported for both, filament, and mucus cells during wound healing, but not for club cells. [52]. In other studies, mechanical wounding with needle pokes resulted in no change in club cell density in yellow perch *Perca flavescens* [25], but increased the club cell size, number, their upward migration, and the levels of rough endoplasmic reticulum, Golgi membranes, and leucocyte incorporation within their cytoplasm [152]. Skorić et al. reported an increased number of ECCs in 75% of the specimens of mirror and scaly carp injured by the attacks of a fish-eating bird [168].

The relationship between predation and immune function is an emerging area of investigation. A recent study showed that chronic exposure to an alarm cue, over a period of 4 years, increased the number of lymphocytes in the blood of alarm cue exposed fish [169]. In another study, the alarm cue showed anti-fungal properties [25]. As previously stated, the alarm cue is purported to be a mixture of several active ingredients, including bacteria [73]; therefore, further studies should parse out the immunostimulatory, antimicrobial ingredient of the alarm cue. While ECCs are hypothesized to be a contributor to alarm substance [4], exposing fathead minnows *Pimephales promelas* to the alarm cue (skin extracts from same or distant fish species) or hypoxanthine-3-N-oxide did not affect ECC density [25]. Exposure to the alarm cue does, however, increase cortisol levels [170]. In previous studies, cortisol has been shown to modulate the immune response [146,171,172,173] and decrease club cell density [146]. Cortisol levels were also increased upon exposure to UV rays [19,20], and UV rays can affect other aspects of fish’s immune systems [174]. Clearly, further studies are needed to establish the mechanistic links and signaling pathways.

Another important, yet under-investigated area is the interaction between sex hormones and ECCs and immune function. In mice and humans, sex hormones can affect the immune response to pathogens [175,176,177]. Testosterone is known to affect the immune system in fish [178,179,180,181]. Male fathead minnows treated with testosterone lose their club cells [157]. In addition, histological examination of skin from fathead minnow females and males showed that breeding males temporarily lose their club cells, which coincides with peak androgen production [182]. Similarly, treatment with 17α-methyltestosterone reduced club cells counts in male and female zebra danios *Danio rerio* [183]. Smith hypothesized that loss of ECCs was because egg-rubbing by nesting males could release alarm cue and repel potential mates [184]. However, testing this hypothesis would require experimental creation of reproductively-active males with ECCs. Moreover, since male zebrafish treated with testosterone lose their ECCs, and zebrafish do not engage in abrasive egg rubbing behavior, attributing seasonal loss of ECCs to abrasive behavior is not the most parsimonious explanation. A plausible alternative explanation for seasonal loss of ECCs during the breeding season is that males reallocate resources from ECCs to nest defense, courtship, and egg care, resulting in severe energetic constraints [185]. A support for energy constraints influencing investment in club cells comes from Wisenden et al.’s study which showed that elevated food ration increased ECC density [158] (but also see [159]).

Previous studies noted the presence of other cell types and intrusions within club cells [152,186,187]. Chia et al.’s study recently followed up on these observations in a detailed image-based analysis enabled by fluorescently tagged markers [73]. They showed that bacteria are transported with mucus into ECCs, and this potentially involve transcytosis or invasion by another cell type such as neutrophils. With caspase-3 based staining, they described invasion as a cellular uptake mechanism for apoptotic cells, distinct from phagocytosis. The biological relevance of such a mechanism has not yet been investigated, and may hold relevance for antigen presentation by club cells, which would allow them to bridge innate and adaptive immune responses in fish.

Therefore, as discussed above, there are several lines of evidence to support the immune function of ECCs: (1) They are strategically located in mid-epidermal layer of the skin, which is exposed to numerous immunomodulators/environmental stressors and forms the first line of defense against pathogens and parasites; (2) they are responsive to many immunomodulators/environmental stressors, including cortisol, pathogens, parasites, UV rays, mechanical injury, heavy metals, testosterone, food ration, etc. (see Table 2); (3) many immunomodulators have been observed inside the ECCs including chondroitin and keratin sulfate, leukocytes, serotonin, mucus, and bacteria. The studies reviewed in Table 2 primarily adopted a histological approach to investigate club cell density and area, and made indirect inferences on its function based on microscopic observations. Although these circumstantial and indirect inferences provide a satisfying explanation to evolutionary ecologists, immunological data demonstrating cellular and molecular mechanisms of immune function for ECCs are completely lacking.

## 4. Future Research Directions

There are multiple open lines of research that would shed light on the ways in which ECCs serve as part of the immune system, and how ECCs may serve as a tool for linking immune function with other life history traits.

### 4.1. Proposed Experimentation for Characterizing Club Cells’s Immune Functions

Accumulating evidence indicates that epidermal club cells are innate immune cells (Figure 2), which may participate in several immune functions such as: (1) Recognition of pathogens or damage-associated ligands through pattern recognition receptors (PRRs). While several PRRs have been identified in fish (see section on innate immune system), ECC-specific gene or protein expression of PRRs has not been pursued; (2) antimicrobial functions of ECCs can be investigated with the help of functional assays such as phagocytic assays, respiratory burst assays, or by their ability to produce antimicrobial peptides and reactive oxygen species; (3) their role in antigen-presentation can be investigated by elucidating cell-specific expression of major histocompatibility complex and co-stimulatory molecules, and by their ability to process an antigen; (4) their potential to activate adaptive immune responses, mediate cell-to-cell communication or influence paracrine interactions could be determined by investigating club cell-specific expression of cytokines, chemokines, and cell growth/differentiation factors; (5) their potential to participate in innate immune memory response could be especially valuable in conferring long-term protection against pathogens, and can be mechanistically investigated by studying epigenetic modifications such as chromatin remodeling, microRNA expression, DNA methylation, and histone modification; (6) their role in production of extracellular matrix components (chondroitin and keratan sulfate positive) or in environmental sensing as a paraneuronal cell (serotonin positive) or as photo-protective cells (calcium-binding protein expression) requires club cell-specific functional assays. While fish-specific tools for gene and protein analysis have greatly expanded in recent years, progress on immune function of club cells has lagged considerably. One major hindrance is that ECC-specific cell-surface markers or cell contents that would be critical in designing flow-cytometry based experiments or for ascertaining their purity in primary cultures are hitherto unknown. Nonetheless, as previously stated, club cells can be clearly identified in histological sections, based on their anatomical location, morphology and staining pattern, and this should enable experiments based on laser-capture microdissection for gene expression or transcriptomic analysis or protein expression using immunohistochemistry.

### 4.2. Ecological Trade-Offs with the Immune System: An Ecoimmunological Perspective on Epidermal Club Cells

Ecoimmunology is the study of the dynamic processes that integrate internal physiological mechanisms regulating immunocompetence, metabolism, energetics, growth, and reproduction with external ecological and evolutionary factors [190,191,192,193]. Life history traits influence one another in that investment in one trait constrains investment in other traits [194]. For example, allocation of resources to somatic growth comes at a cost to reproductive output [195]. Trade-offs arise from ecological interactions among competitors (e.g., optimal foraging) and between prey and their predators (e.g., risk-sensitive foraging [196], and risk-sensitive sexual displays [197,198,199]). Parasites and pathogens exert selection on hosts in part by forcing the host to reallocate resources to mounting an immune defense and/or production of parasite/pathogen propagules. For example, guppies *Poecilia reticulata* infected with the ectoparasitic monogene *Gyrodactylus* are forced to reallocate carotenoids (pigments) used for sexual displays to immune responses, resulting in reduced intensity of orange pigment displays [200,201]. Similarly, parasites that penetrate the skin, such as the cercariae of trematodes and fungal spores can reduce fat storage and thus increase the likelihood of tradeoffs [202] (Figure 3).

Immune function is highly plastic, which is critical for a host’s ability to combat a diverse array of pathogens. However, activation of the immune system’s components demands extensive resources from the host, and certain components of immune system are more costly than others. For example, pre-formed elements of innate immunity, like mucus and antimicrobial peptides, which are critical in providing immediate defense is likely to require less resources than the ones involving systemic activation and turnover of cellular components (e.g., phagocytic macrophages). Similarly, the cell-mediated (T lymphocytic) immune responses, which are essential for tackling intracellular pathogens, are likely to be more metabolically expensive than antibody-mediated humoral immune response against extracellular pathogens [203,204]. The elements of the immune system that are costlier to maintain are more likely to be affected by environmental stressors [190]. These elements of immune system can be regulated independently within an individual within the constraints of prevailing local conditions, and can also vary tremendously between species and taxa. This complexity makes demonstrating and measuring tradeoffs between investment in immune function versus other functions a major challenge for the field of ecoimmunology [205].

## 5. ECCs as a Tool for Ecoimmunological Studies

We propose that epidermal club cells (ECCs) present a candidate indicator of ecoimmunological trade-offs. If ECCs are indeed involved as a first line of defense against epidermal injury, or secondary bacterial infections caused by these wounds, then we can make testable predictions about how ECC density might covary in time and space with temporal and spatial variation in food availability, or seasonal and local variation in parasite density. Preliminary data indicate that there is large intraspecific variation in ECCs among sites [206] and likely over seasonal temporal scales as well. From the perspective of nutrient availability, ECC density should be greatest in the summer when food is plentiful and lowest in the winter (in temperate climes) when food is scarce, and generally greater in eutrophic systems than in oligotrophic systems. Ontogenetically, ECCs should be reduced during times of exceptional energetic demands for other purposes such as rapid growth [21], nest defense, and egg care [157,184,207], and other activities such as sustained migration.

From the perspective of immune function, ECC density should respond and/or be activated at times and in places where encounter rates with pathogens are greatest. ECC numbers should be more active in complex biological communities that support a diversity of parasites and pathogens than in simple or isolated communities. For example, parasites such as trematode flatworms require specific gastropod and fish intermediate hosts, and are trophically transmitted from fish to avian final hosts. Complex life cycles with multiple hosts require complex communities. ECC production should also track seasonal variation in ceracarial release and other temporally variable pathogens. We predict that populations or individuals at the edge of their geographic distribution, or those expanding their range should have high densities of ECCs stimulated by encounters with novel parasites and pathogens. For example, invasive species should have highly active and responsive ECCs either as a consequence of expanding into new areas, or because invasive species may be pre-adapted for vicariance because of naturally high level of ECCs. Alternatively, changes in host–pathogen interactions can occur by shifting distributional ranges of one or the other due to the effects of climate change.

From the perspective of phylogeny, fishes that naturally occur at high densities, such as shoaling species, have relatively high risk of horizontal transmission of pathogens among group members and should invest more heavily in ECCs than nonsocial solitary species. Notably, cyprinids and characins within the ostariophysi are virtually all obligate schooling species.

The best data for a systematic test of these ideas to date come from fathead minnows collected from four sites in Saskatchewan [206]. These data indicated marked intraspecific variation in ECC number per mm (of a histological preparation of epidermal tissue) across four populations and even among collection loci within one of the sites. There was also a difference in the overall thickness of the epidermis, which meant that differences in the number of ECCs per mm were achieved not so much by changing ECC density but by changes in epidermal thickness. When minnows from these field sites were held in the lab under standard conditions, the number of ECCs per mm converged by the end of a 28-day observation period [206]. These data indicate that population differences in ECC number dynamically respond to changes in environmental conditions, and population differences in ECCs likely reflect conditions present at each site, or sub-site. In support of this hypothesis, Snider [27] recently reported significant differences in ECC densities between wild-caught and lab-reared fathead minnows.

If these hypotheses are ultimately supported by empirical observations, then a second line of research might explore why ECCs are not more common than they seem to be. For example, all fishes are exposed to parasites of one form or another, and there are many schooling species that apparently lack ECCs. What role do phylogenetic constraints play in the distribution of ECCs among fishes? How do the immune defenses of fishes that lack ECCs compare to those that have them? What is the role of ECCs in marine fishes? Knowledge of the mechanisms and pathways ECCs in the immune system of fishes will shed light on many of these questions.

The ecoimmunology of epidermal club cells (ECCs) is ripe for exploration. Even a basic understanding of the immune function of ECCs would inform the evolutionary ecology of these cells, the fishes that do and do not have them, and contribute significantly to a very large literature on the role of injury-released chemical cues in mediating predator–prey interactions. In the process, an ecoimmunological approach would expand our understanding of the evolution of immunological responses in fishes, and in vertebrates generally.

## Figures and Tables

**Figure 1 ijms-22-01440-f001:**
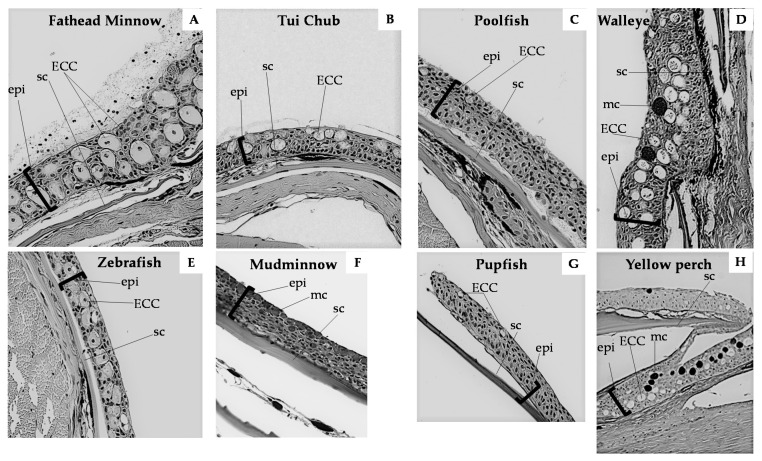
Histological preparations of epidermal tissues from representative fish species showing variation in size, position, and characteristics of epidermal club cells, periodic acid–Schiff (PAS) stain, 400×. (**A**) Cypriniformes, fathead minnow *Pimephales promelas*, note that one ECC in this image is binucleate, (**B**) Cypriniformes, zebrafish *Danio rerio*, (**C**) Cypriniformes, Hot Creek tui chub *Siphateles bicolor* ssp., (**D**) Esociformes mudminnow *Umbra limi*, (**E**) Cyprinodontiformes, Pahrump poolfish *Empetrichthys latos*, (**F**) Cyprinodontiformes, Amargosa pupfish *Cyprinodon nevadensis amargosae*, (**G**) Perciformes, walleye *Sander vitreus*, (**H**) Perciformes, yellow perch *Perca flavescens*. ECC, epidermal club cell; epi, epidermal thickness; sc, scale; mc, mucus cell.

**Figure 2 ijms-22-01440-f002:**
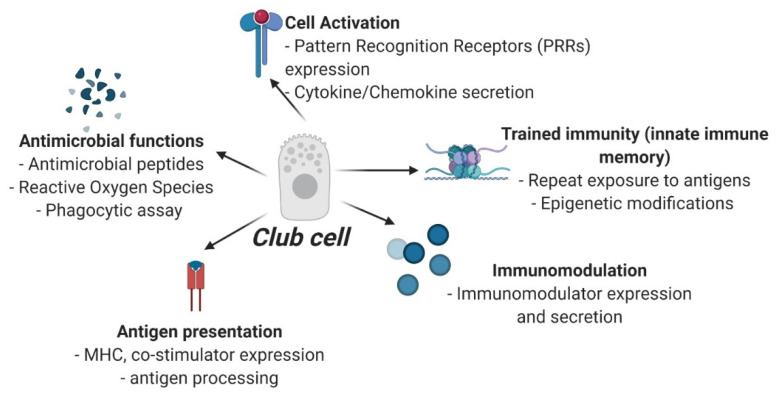
Proposed experiments for in-depth characterization of club cell’s immune function. Schematic created using www.biorender.com.

**Figure 3 ijms-22-01440-f003:**
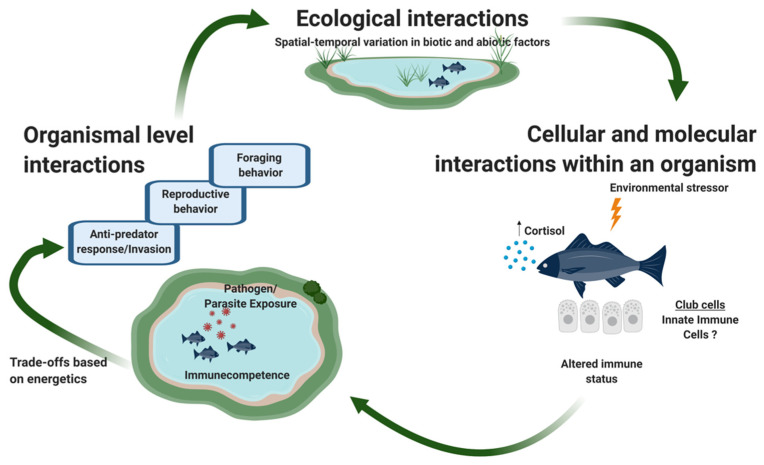
An ecoimmunological perspective on epidermal club cells (ECCs). An ecoimmunological perspective integrates external environmental interactions with internal cellular and molecular immunological dynamics. ECCs may benefit an individual by serving as a component of the innate immune system, although the specific functions they serve are poorly understood and may vary widely among species. ECCs respond to environmental stressors, such as pathogens, UV rays, mechanical damage epidermis, seasonal changes (e.g., breeding season), etc. These stressors can elevate cortisol levels, which can in turn increase ECC density, and alter an organism’s immunological state. The immune status of an individual in turn determines its competence to fight pathogens and parasites in the environment. The cost of activating immune system components has trade-offs with other demands on an organism and can influence its anti-predator response, invasive capabilities—which require exploration of new territories, environments, pathogens, and parasites, foraging activity, and reproductive success. These trade-offs will vary spatially and temporally in response to geographic and seasonal variation in abiotic and biotic factors. Schematic created using www.biorender.com.

**Table 1 ijms-22-01440-t001:** Phylogenetic distribution of epidermal club cells in fishes.

Superclass	Class	Subclass	Infraclass/Division	Superorder	Order	Common Name	ECCs
Agnatha						Lamprey	Skein cells, which are distinct from club cells [31]
Gnathostoma	Chondrichthyes					Sharks, rays	Skein cells [32]
	Osteichthyes	Sarcopterygii				Coelacanth	Absent [33]
		Actinopterygii	Chondrostei		Polypteriformes	Reed fish	Present [34]
			Teleostei	Elopomorpha	Anguilliformes	Eels	Present [35]
				Ostariophysi	Gymnotiformes	Electric fishes	Secondary loss of ECCs due to electric sense [4,36]
					Gonorynchiformes	Milkfish	All possess ECCs [37]
					Siluriformes	Catfish	All possess ECCs [4,34,38,39,40,41,42,43,44,45,46]
					Characiformes	Characins	Most possess ECCs [4,47,48,49,50], absent in others [4]
					Cypriniformes	Minnows, zebrafish	All possess ECCs [4,24,41,51,52]
				Protacanthopterygii	Salmoniformes	Salmon, trout, charr	Possess ECCs [53], absent in others [24,54]
					Esociformes	Mudminnows, pike	Absent [55]
				Paracanthopterygii	Ophidiiformes	Pearlfish	Possess ECCs [44]
					Gadiformes	Cod	Possess ECCs [37]
				Protacanthopterygii *	Cyprinodontiformes	Pupfish, poolfish, mosquitofish	ECCs in low density [27,38], others lack them [56]
					Perciformes	Perch, walleye, darters, cichlids, sunfish	Some species have them, others lack them [25,50,55,57]
					Gasterosteiformes	Stickleback	Absent [4,58]
					Gobiiformes	Gobies	Some species have ECCs [59,60]
					Tetraodontiformes	Pufferfish	Some species have ECCs [61]

* Not all orders are listed to conserve space. Orders omitted from the table that do not have ECCs according to Pfeiffer [4]: Beloniformes (also [34]), Carangiformes, Anabantiformes, Blenniformes, Callionymiformes, Gobioesociformes, Labriformes, Mugiliformes, Ovalentaria, Pleuronectiformes, Scombriformes, Scorpaeniformes [62], Trachiniformes.

**Table 2 ijms-22-01440-t002:** Summary of environmental stressors and their effect on epidermal club cells in fishes.

Environmental Stressor	Club Cell Measurement and Observed Effect	Fish Species	Reference
Predation			
(a) Skin extracts	Club cell density—no effect	Fathead minnows	Chivers et al., 2007 [25]
(b) Hypoxanthine-3-N-oxide	Club cell density—no effect	Fathead minnows	Chivers et al., 2007 [25]
(c) Cortisol	Club cell density—decrease, individual club cell area—unchanged	Fathead minnows	Halbgewachs et al., 2009 [146]
(d) Euthanizing agent—MS222 or Aquacalm	No effect	Fathead minnows	Manek et al., 2014 [20]
Parasites			
(a) *Uvulifer ambloplites* (black spot disease)	Club cell density—positive correlation	Yellow perch	Chivers et al., 2007 [25]
(b) Metacercariae of the trematode (*Teleorchis* sp.)	Club cell density—increase	Fathead minnows	Chivers et al., 2007 [25]
(c) Cercariae of the trematode (*Ornithodiplostomum* spp.)	Club cell density—no effect	Fathead minnows	James et al., 2009 [160]
(d) *Henneguya multiplasmodialis*	Observational study—club cells in gills were found in close contact with the parasite	Catfish (*Pseudoplatystoma* spp.)	Adriano et al., 2012 [147]
Infections			
(a) *Saprolegnia ferax*, *S. parasitica*	Club cell numbers—increase	Fathead minnows	Chivers et al., 2007 [25]
(b) *S. ferax* in presence of Cadmium (Cd)	Club cell numbers—increase with no or low dose Cd but remain unchanged with high dose Cd	Fathead minnows	Chivers et al., 2007 [25]
(c) *S. ferax* and *S. parasitica*	Smaller club cells with high (but not low) concentrations of the pathogens	Fathead minnows	Pollock et al., 2012 [148]
(d) *Aeromonas hydrophila*	Club cell density—decrease Club cell degeneration following extensive vacuolization	Indian carp, *Labeo rohita*	Srivastava et al., 2020 [149]
(e) Various bacteria	Club cell degeneration following extensive vacuolization—positive association	Catfish (*Clarias gariepinus*)	El-Sayyada et al., 2010 [188]
Radiations and light			
(a) Ultraviolet (UV)-B radiations	Club cells—hypertrophy and hyperplasia	Razorback suckers	Blazer et al., 1997 [24]
(b) UV rays	Club cell density—decrease	Fathead minnows	Manek et al., 2012 [19]
(c) UV rays	Club cell density—decrease, club cell area—no effect	Fathead minnows	Manek et al., 2014 [20]
(d) UV-A rays	Club cell hypertrophy	Catfish (*Clarias gariepinus*)	Sayed et al., 2013 [151]
(e) UV-rays with or without dissolved organic carbon (DOC)	Club cell density decrease, and area is unchanged with UV rays, club cell density unchanged with UV rays in presence of DOC	Fathead minnows	Manek et al., 2014 [150]
(f) White light with 34.8% of blue light emission and white-blue light with 54.6% of blue light emission	Club cell numbers increase, expression and colocalization of calbindin and calretinin proteins in club cells	Zebrafish, *Danio rerio*	Lauriano et al., 2020 [79]
Water pollutants			
(a) Acidified water (pH 5 or 6)	Club cell size, number, showed upward migration towards the skin surface, rough endoplasmic reticulum, Golgi membranes, and leucocyte incorporation—increase	Juvenile carp, *Cyprinus carpio*	Iger et al., 1994 [152], [186], Iger et al., 1988 [153]
(b) Brackish water			
(c) Chicken Manure			
(d) Detergent—sodium dodecyl sulphate	Club cell degeneration following extensive vacuolization	Catfish (*Clarias batrachus*)	Mittal et al., 1994 [154]
(e) Detergent—linear alkylbenzene sulfonate	Club cell density—decrease	*Prochilodus lineatus*	Alves et al., 2016 [189]
(f) Azo dye—Eriochrome black T.	Club cell degeneration following extensive vacuolization, club cell density—increase	Carp, *Labeo rohita*	Srivastava et al., 2017 [155]
Heavy metals			
(a) Cadmium	Club cell size, number, upward migration, rough endoplasmic reticulum, Golgi membranes, and leucocyte incorporation—increase	Juvenile carp, *Cyprinus carpio*	Iger et al., 1994 [152]
(b) Lead			
(c) Cadmium (+UV rays)	No effect	Fathead minnows	Manek et al., 2014 [20]
(d) Copper	Club cells—elongated, showed upward migration towards the skin surface, cytoplasm contained extensive rough endoplasmic reticulum, Golgi system and free ribosomes, lysosomes and phagosomes present, newly differentiated cells located in the mid-epidermis	Juvenile carp, *Cyprinus carpio*	Iger et al., 1994 [156]
Mechanical injury			
(a) Needle poke	Club cell numbers—no effect	Yellow perch	Chivers et al., 2007 [25]
(b) Wounding	Club cell size, number, upward migration, rough endoplasmic reticulum, Golgi membranes, and leucocyte incorporation—increase	Juvenile carp, *Cyprinus carpio*	Iger et al., 1994 [152]
(c) Experimental wounds	Similar numbers in wounded and normal tissue, last cell type to differentiate from filament cells	Mirror carp, *Cyprinus carpio*	Iger et al., 1990 [52]
(d) Wounds caused by aquatic bird—great cormorant	Increased club cells in 75% of the specimens	Mirror and scaly carp, *Cyprinus carpio*	Skorić et al., 2012 [168]
Sex hormones			
(a) Testosterone	Club cell density—decrease	Fathead minnows	Smith, 1973 [157]
(b) 17α-methyltestosterone	Club cell numbers—decrease	Zebra danios	Smith, 1986 [183]
(c) Breeding males	Temporary loss of club cells	Fathead minnows	Smith et al., 1974 [182]
Food ration			
(a) High food ration	Club cell density increase	Fathead minnows	Wisenden et al., 1997 [158]
(b) Food ration	Club cell density—no effect	Catfish (*Pseudoplatystoma corruscans*)	Barreto et al., 2012 [159]

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
