# Peer review of "Epidermal Club Cells in Fishes: A Case for Ecoimmunological Analysis"

_ijms, 2021, doi:10.3390/ijms22031440_

Round 1
Reviewer 1 Report
Brief summary
The manuscript "Epidermal club cells in Ostariophysan fishes: a case for ecoimmunological analysis?” seeks to understand the evolutionary reasons for investment in Epidermal club cells (ECCs) by different fish taxa. For this, the authors reviewed the phylogenetic distribution of ECCs and their histochemical properties. I believe this study is a contribution to assist in understanding the response of ECCs to several factors, including physiological, environmental and biological. Overall, the provided information is a good resource; however, unfortunately the authors have not fully developed some topics of the manuscript and I have decided that the manuscript needs major revision.
Broad comments to authors:
Overall, I recommend clarifying some aspects:
- The title makes the reader imagine that the research is focused on ECCs in Ostariophysan fish, however, the objectives are not aligned with it, as well as the development. It seems to be much broader.
- Contextualization of the introduction and objective (for example, some aspects presented in the objectives and results were not widely contextualized in the introduction); of the methods in order to allow results interpretation; and discussion, which should be further expanded and justified.
- It’s unclear what questions you are asking, why it’s interesting, why ECCs in Ostariophysan fishes can be interesting to be researched in an ecoimmunological context. In the introduction, for example, leave your goals in topic style (i, ii, iii ...) this can facilitate understanding.
- I believe that the current structure of the manuscript makes it difficult to understand, I think. I think the manuscript needs a much clearer structure that is closely linked to questions (which need to be defined much more clearly, please see earlier comments), and the results of those questions. There are lots of information and ideas, but I’m struggling to see how it all fits together, and how it fits with the data collected, and what the story is.
For example, it is not possible to see in the text the method used to select the papers that addressed ECCs in Ostariophysan fish. In what period (month / year) was the review carried out? Was it done on any database? Were there any temporal and spatial cuts? Was any term used to search for the works used in this review? For example, 'Epidermal club cells'?
These are the main problems I found in the manuscript, but there are many other minor mistakes or suggestions I indicated below, and I hope they may help the authors when reviewing their work. The detailed suggestions follow below.
Specific comments to authors:
Lines 19-22: Perhaps it would be helpful standardize the objective described here with that mentioned in the introduction.
Line 26: ECC density varies widely among and within fish populations? has this been tested / analyzed?
Lines 29-30: I’m not quite sure why this is noteworthy. This seems to be largely just re-stating the approach you’ve taken.
Line 30-31: perhaps these should be being taken into account in this study?
Lines 32-34: many keywords, more than 10. Maybe remove one.
Line 44: remove English-language. Not necessary.
Line 150: the table caption should be better detailed. For example, phylogenetic distribution of cells in the epidermal club in which?
Line 151: it is not possible to understand the note. Where can you see the asterisk in the table? Furthermore, what figure is that mentioned? This is confusing in the text.
Line 177: Pimephales promelas in italics.
Line 178: Danio rerio in italics.
Line 179: Siphateles bicolor in italics.
Line 180: Umbra limi in italics.
Line 180: Empetrichthys latos in italics.
Line 181: Cyprinodon nevadensis in italics.
Line 182: Sander vitreus and Perca flavescens in italics.
Line 223: précis?
Table 2: the table needs to be revised, several minor errors, for example: formatting; 'sp' without point; common names without the scientific [e.g., razorback suckers (Xyrauchen texanus); fathead minnow (Pimephales promelas); yellow perch (Lose flavescens)]; sometimes it cites the common name and then the specific one in parentheses, sometimes it doesn't.
Lines 689-695: I would recommend adding information in Conclusions section to tie things together where you highlight the main findings of the study and also the implications of these.
Supplementary: Would it be possible to provide a supplementary table with all the articles analyzed for this review?
Author Response
Reviewer 1
Brief summary
The manuscript "Epidermal club cells in Ostariophysan fishes: a case for ecoimmunological analysis?” seeks to understand the evolutionary reasons for investment in Epidermal club cells (ECCs) by different fish taxa. For this, the authors reviewed the phylogenetic distribution of ECCs and their histochemical properties. I believe this study is a contribution to assist in understanding the response of ECCs to several factors, including physiological, environmental and biological. Overall, the provided information is a good resource; however, unfortunately the authors have not fully developed some topics of the manuscript and I have decided that the manuscript needs major revision.
Broad comments to authors:
Overall, I recommend clarifying some aspects:
- The title makes the reader imagine that the research is focused on ECCs in Ostariophysan fish, however, the objectives are not aligned with it, as well as the development. It seems to be much broader.
>>> We agree with this comment and have dropped the reference to Ostariophysi from the title.
- Contextualization of the introduction and objective (for example, some aspects presented in the objectives and results were not widely contextualized in the introduction); of the methods in order to allow results interpretation; and discussion, which should be further expanded and justified.
>>> We were not sure what this reviewer meant by “contextualization”. We feel that the document flows well from one section to the next. We were not able to revise the manuscript in response to this comment.
- It’s unclear what questions you are asking, why it’s interesting, why ECCs in Ostariophysan fishes can be interesting to be researched in an ecoimmunological context. In the introduction, for example, leave your goals in topic style (i, ii, iii ...) this can facilitate understanding.
>>> We are puzzled by this comment because we think we clearly present a historical record for how ECCs came to be thought of as the source of alarm cues, how evolutionary biologists found fault with that logic, and the emerging consensus among ecologists that ECCs have an immune function, but little awareness among immunologists of this literature. The role of this review is bring these two communities of scientists together.
- I believe that the current structure of the manuscript makes it difficult to understand, I think. I think the manuscript needs a much clearer structure that is closely linked to questions (which need to be defined much more clearly, please see earlier comments), and the results of those questions. There are lots of information and ideas, but I’m struggling to see how it all fits together, and how it fits with the data collected, and what the story is.
>>> The structure is logical (to us): (1) What is the historical context of ECCs? (2) What is the phylogenetic distribution of ECCs? (3) What are the histochemical properties of ECCs that might infer function? (4) A brief review of the innate and adaptive immune system in fishes to provide context (comment #2?) for making the case that ECCs play a role in immune defense. (5) Future research ideas for advancing these ideas, and (6) a theoretical framework provided by ecoimmunology to integrate immunological mechanisms with evolutionary ecology.
For example, it is not possible to see in the text the method used to select the papers that addressed ECCs in Ostariophysan fish. In what period (month / year) was the review carried out? Was it done on any database? Were there any temporal and spatial cuts? Was any term used to search for the works used in this review? For example, 'Epidermal club cells'?
>>> This is a review of all existing knowledge, which required multiple searches with a variety of key word criteria in every data base we could think of, and subsequent follow up on references cited in the papers we uncovered. It appears as if the reviewer is thinking of the style of literature review where the “data” are defined as X number of papers that were the result of searches using Y search criteria on Z databases. This approach to a literature review would not have served our needs very well because we synthesized to disparate literatures and a literature that goes back to the 1930s.
These are the main problems I found in the manuscript, but there are many other minor mistakes or suggestions I indicated below, and I hope they may help the authors when reviewing their work. The detailed suggestions follow below.
Specific comments to authors:
Lines 19-22: Perhaps it would be helpful standardize the objective described here with that mentioned in the introduction.
Line 26: ECC density varies widely among and within fish populations? has this been tested / analyzed?
>>> Yes, by Manek et al. 2013, as described in the final section of the review.
Lines 29-30: I’m not quite sure why this is noteworthy. This seems to be largely just re-stating the approach you’ve taken.
>>> The reviewer is referring to this sentence: Here, we review the case for ECC immune function, immune functions in fishes generally, and encourage future work describing the precise role of ECCs in the immune system of fishes.
Yes, we followed the old adage of telling them what you’re going to tell them, tell them, tell them what you told them. We feel this is useful and kept it in.
Line 30-31: perhaps these should be being taken into account in this study?
>>> We think this comment refers to describing the immune functions in fishes generally and encouraging future research on the precise role of ECCs in the immune functions of fishes. We covered those topics in our review.
Lines 32-34: many keywords, more than 10. Maybe remove one.
>>> We now list only mucosal immune system; epidermal club cells; ostariophysi; ecoimmunology
Line 44: remove English-language. Not necessary.
>>> removed
Line 150: the table caption should be better detailed. For example, phylogenetic distribution of cells in the epidermal club in which?
>>> We are confused by this comment. We think the table heading “Phylogenetic distribution of epidermal club cells” is clear.
Line 151: it is not possible to understand the note. Where can you see the asterisk in the table? Furthermore, what figure is that mentioned? This is confusing in the text.
>>> We have added the asterisk to the table that links to the footnote. We thank the reviewer for catching this oversight.
Line 177: Pimephales promelas in italics.
>>> added
Line 178: Danio rerio in italics.
>>> added
Line 179: Siphateles bicolor in italics.
>>> added
Line 180: Umbra limi in italics.
>>> added
Line 180: Empetrichthys latos in italics.
>>> added
Line 181: Cyprinodon nevadensis in italics.
>>> added
Line 182: Sander vitreus and Perca flavescens in italics
>>> added
Line 223: précis?
>>> We reworded the heading to be ostentatious
Table 2: the table needs to be revised, several minor errors, for example: formatting; 'sp' without point; common names without the scientific [e.g., razorback suckers (Xyrauchen texanus); fathead minnow (Pimephales promelas); yellow perch (Lose flavescens)]; sometimes it cites the common name and then the specific one in parentheses, sometimes it doesn't.
>>> We have revised the table to provide this information
Lines 689-695: I would recommend adding information in Conclusions section to tie things together where you highlight the main findings of the study and also the implications of these.
>>> The reviewer has done us a great service in pointing out that the landing was a bit muddled. We have several concluding paragraphs; one for future research on immune function, one for future research on life history trade-offs, and the final paragraph that integrates the two fields into an ecoimmunological framework. We no longer have a paragraph titled “Conclusions”.
Supplementary: Would it be possible to provide a supplementary table with all the articles analyzed for this review?
>>> We cite our sources in the references section, and also in tables 1 and 2.

Reviewer 2 Report
I believe that the main claims of your review:
The main claims of the paper:
- Epidermal club cells (ECCs) have been extensively studied in the context predator-prey ecology, because they are the presumed source of chemical alarm cues released during predator attacks.
- The apparent absence of ECCs in many groups is likely an artifact of low sampling effort.
- ECCs may be a contributor to, but not the sole source of alarm cue.
- Individual fish would not realize a fitness benefit for investing in ECCs and thus their maintenance must be explained by some other adaptive function, which benefits the sender.
- An alternative hypothesis for the function for ECCs is a role in immune defence further supported by accumulating evidence indicating that epidermal club cells are in fact innate immune cells.
- Immunological data demonstrating cellular and molecular mechanisms of immune function for ECCs are completely lacking.
- The concept of immunological memory has long been associated with adaptive immune system. However, the concept of trained immunity in innate immune system and its role in defense against skin pathogens is relatively newer.
- Understanding the immunological processes involved in wound-healing is important to design evidence-based diagnostics and therapeutics.
Are valid.
The paper is well written and well-structured.
Author Response
Reviewer 2
I believe that the main claims of your review:
The main claims of the paper:
- Epidermal club cells (ECCs) have been extensively studied in the context predator-prey ecology, because they are the presumed source of chemical alarm cues released during predator attacks.
- The apparent absence of ECCs in many groups is likely an artifact of low sampling effort.
- ECCs may be a contributor to, but not the sole source of alarm cue.
- Individual fish would not realize a fitness benefit for investing in ECCs and thus their maintenance must be explained by some other adaptive function, which benefits the sender.
- An alternative hypothesis for the function for ECCs is a role in immune defence further supported by accumulating evidence indicating that epidermal club cells are in fact innate immune cells.
- Immunological data demonstrating cellular and molecular mechanisms of immune function for ECCs are completely lacking.
- The concept of immunological memory has long been associated with adaptive immune system. However, the concept of trained immunity in innate immune system and its role in defense against skin pathogens is relatively newer.
- Understanding the immunological processes involved in wound-healing is important to design evidence-based diagnostics and therapeutics.
Are valid.
The paper is well written and well-structured.
>>> We agree with comments from this review.

Round 2
Reviewer 1 Report
The manuscript "Epidermal club cells in Ostariophysan fishes: a case for ecoimmunological analysis?” seeks to understand the evolutionary reasons for investment in Epidermal club cells (ECCs) by different fish taxa. For this, the authors reviewed the phylogenetic distribution of ECCs and their histochemical properties. I believe this study is a contribution to assist in understanding the response of ECCs to several factors, including physiological, environmental and biological. Overall, the provided information is a good resource. The authors have made efforts to amend the manuscript based upon the original comments. The changes made by the authors improve the manuscript. On balance, I believe this manuscript is now suitable for publication.
Some details:
Line 179: spp. non italic
Line 150: the table caption should be better detailed. Captions must be self-explanatory. So it is best to make it clear that it is ECCs in fish.
Suggestion:
Table 1. Phylogenetic distribution of epidermal club cells in fishes.
Line 550: Captions must be self-explanatory. So it is best to make it clear that it is ECCs in fish.
Suggestion:
Table 2. Summary of environmental stressors and their effect on epidermal club cells in fishes.
In table 2: spp. non italic. See Pseudoplatystoma spp.